# Batch-Instance Normalization for Adaptively Style-Invariant Neural Networks

**Hyeonseob Nam**
Lunit Inc.
Seoul, South Korea
hsnam@lunit.io

**Hyo-Eun Kim**
Lunit Inc.
Seoul, South Korea
hekim@lunit.io

## Abstract

Real-world image recognition is often challenged by the variability of visual styles including object textures, lighting conditions, filter effects, etc. Although these variations have been deemed to be implicitly handled by more training data and deeper networks, recent advances in image style transfer suggest that it is also possible to explicitly manipulate the style information. Extending this idea to general visual recognition problems, we present Batch-Instance Normalization (BIN) to explicitly normalize unnecessary styles from images. Considering certain style features play an essential role in discriminative tasks, BIN learns to selectively normalize only disturbing styles while preserving useful styles. The proposed normalization module is easily incorporated into existing network architectures such as Residual Networks, and surprisingly improves the recognition performance in various scenarios. Furthermore, experiments verify that BIN effectively adapts to completely different tasks like object classification and style transfer, by controlling the trade-off between preserving and removing style variations. BIN can be implemented with only a few lines of code using popular deep learning frameworks.[1]

## 1 Introduction

Information described by an image generally consists of spatial and style information, such as object shape and texture, respectively. While the spatial information usually represents the key contents of the image, the style information often involves extraneous details that complicate recognition tasks. Typical examples include the discrepancies in object textures caused by different camera settings and light conditions, which disturb object classification. Despite the quantum leap of deep learning in computer vision, visual recognition in real-world applications still suffers from these style variations inherent in images.

Moderating the variability of image styles has been studied recently to enhance the quality of images generated from neural networks. Instance Normalization (IN) [22] is a representative approach which was introduced to discard instance-specific contrast information from an image during style transfer. Inspired by this, Huang *et al*. [10] provided a rational interpretation that IN performs a form of style normalization, showing simply adjusting the feature statistics—namely the mean and variance—of a generator network can control the style of the generated image. Due to this nature, IN has been widely adopted as an alternative to Batch Normalization (BN) in style transfer [14, 3] and generative adversarial networks (GANs) [13, 27].

It is a reasonable assumption that IN would be beneficial not only in generative tasks but also in discriminative tasks for addressing unnecessary style variations. However, directly applying IN to a classification problem degrades the performance [22], probably because styles often serve as useful

discriminative features for classification. Note that IN dilutes the information carried by the global statistics of feature responses while leaving their spatial configuration only, which can be undesirable depending on the task at hand and the information encoded by a feature map. For example, even though the global brightness of an image is commonly irrelevant to object classification, it becomes one of the key features in weather or time prediction. Similarly, the texture of a cloth may confuse classifying clothing categories (shirts vs. skirts), but it is crucial for classifying fashion attributes (spotted vs. striped). In short, normalizing styles in a neural network needs to be investigated with a careful consideration.

In this paper we propose *Batch-Instance Normalization* (BIN) to normalize the styles adaptively to the task and selectively to individual feature maps. It learns to control how much of the style information is propagated through each channel of features leveraging a learnable gate parameter. If the style associated with a feature map is irrelevant to or disturbs the task, BIN closes the gate to suppress the style using IN. If the style carries important information to the task, on the other hand, BIN opens the gate to preserve the style though BN. This simple collaboration of two normalization methods can be directly incorporated into existing BN- or IN-based models, requiring only a few additional parameters. Our extensive experiments show that the proposed approach surprisingly outperforms BN in general object classification and multi-domain problems, and it also successfully substitutes IN in style transfer networks.

## 2 Related Work

**Style manipulation.**   Manipulating image styles or textures has been recently studied along with the finding that the feature statistics of a convolutional neural network effectively capture the style of an image. Cimpoi *et al*. [2] exploited the Fisher-Vector representation built on convolutional features, which encodes their high-order statistics, to perform texture recognition and segmentation. Gatys *et al*. [5] addressed texture synthesis by matching the second-order statistics between the source and generated images based on the Gram matrices of feature maps, which has been extended to neural style transfer algorithms [6, 14, 3]. Huang *et al*. [10] further showed that aligning only the first-order statistics, *i.e.*, the mean and variance, can also efficiently transfer image styles. Note that all the above approaches focused on the tasks directly related to image styles—texture recognition, texture synthesis, style transfer, etc. We expand the idea of manipulating style information to general image recognition problems to resolve impeditive style variations and facilitate training.

**Normalization.**   Batch normalization [12] (BN) has become a standard ingredient in constructing a deep neural network, which normalizes internal activations using the statistics computed over the examples in a minibatch. Several variants of BN such as batch renormalization [11], weight normalization [19], layer normalization [1], and group normalization [24] have been developed mainly to reduce the minibatch dependencies inherent in BN. Instance normalization (IN) [22] exhibits another property of normalizing image styles by adjusting per-instance feature statistics, which is further investigated by conditional instance normalization [3] and adaptive instance normalization [10]. The effectiveness of IN, however, has been restricted to image generation tasks such as style transfer and image-to-image translation, because IN may dilute discriminative information that is essential for general visual recognition. To overcome this, our proposed normalization technique combines the advantages of BN and IN by selectively maintaining or normalizing style information, which benefits a wide range of applications.

## 3 Batch-Instance Normalization

The style of an image is commonly interpreted as the information irrelevant to spatial configuration, which is known to be captured by spatial summary statistics of feature responses in a deep convolutional network [5, 6, 14, 10]. Inspired by the work of Huang *et al*. [10], we assume the first-order statistics—the mean and variance—of a convolutional feature map encodes a certain style attribute. In other words, the information carried by each feature map can be divided into two components: a style (the mean and variance of activations) and shape (the spatial configuration of activations).

From this point of view, instance normalization (IN) can be considered as normalizing the style of each feature map while maintaining the shape only. Although it may help to reduce undesirable style variation, it may also cause severe loss of information if a style itself carries an essential feature for

the task (*e.g.*, relevant to class labels). Batch normalization (BN), on the other hand, normalizes feature responses in a minibatch level, which preserves the instance-level style variation unless the batch size is too small. However, it lacks the ability to address the style inconsistency that complicates the problem. Thus, we aim to allow BN and IN to complement each other in order to selectively preserve or normalize the style encoded by each feature map.

Let $\mathbf{x} \in \mathbb{R}^{N \times C \times H \times W}$ be an input minibatch to a certain layer and $x_{nchw}$ denotes its $nchw$-th element, where $h$ and $w$ indicate the spatial location, $c$ is the channel index, and $n$ is the index of the example in the minibatch. BN normalizes each channel of features using the mean and variance computed over the minibatch:

$$
\begin{aligned}
\hat{x}_{nchw}^{(B)} &= \frac{x_{nchw} - \mu_c^{(B)}}{\sqrt{\sigma_c^{2(B)} + \epsilon}}, \\
\mu_c^{(B)} &= \frac{1}{NHW} \sum_N \sum_H \sum_W x_{nchw}, \\
\sigma_c^{2(B)} &= \frac{1}{NHW} \sum_N \sum_H \sum_W \left( x_{nchw} - \mu_c^{(B)} \right)^2,
\end{aligned}
\tag{1}
$$

where $\hat{\mathbf{x}}^{(B)} = \{\hat{x}_{nchw}^{(B)}\}$ is the batch-normalized response. On the other hand, IN normalizes each example in the minibatch independently using per-instance feature statistics:

$$
\begin{aligned}
\hat{x}_{nchw}^{(I)} &= \frac{x_{nchw} - \mu_{nc}^{(I)}}{\sqrt{\sigma_{nc}^{2(I)} + \epsilon}}, \\
\mu_{nc}^{(I)} &= \frac{1}{HW} \sum_H \sum_W x_{nchw}, \\
\sigma_{nc}^{2(I)} &= \frac{1}{HW} \sum_H \sum_W \left( x_{nchw} - \mu_{nc}^{(I)} \right)^2,
\end{aligned}
\tag{2}
$$

where $\hat{\mathbf{x}}^{(I)} = \{\hat{x}_{nchw}^{(I)}\}$ is the instance-normalized response. Note that the style difference between examples is retained in $\hat{\mathbf{x}}^{(B)}$ but removed in $\hat{\mathbf{x}}^{(I)}$. Our goal is to adaptively balance $\hat{\mathbf{x}}^{(B)}$ and $\hat{\mathbf{x}}^{(I)}$ for each channel so that an important style attribute is preserved while a disturbing one is normalized. Batch-Instance Normalization (BIN) achieves this by introducing additional learnable parameters $\rho \in [0, 1]^C$:

$$
\mathbf{y} = \left( \rho \cdot \hat{\mathbf{x}}^{(B)} + (1 - \rho) \cdot \hat{\mathbf{x}}^{(I)} \right) \cdot \gamma + \beta,
\tag{3}
$$

where $\gamma, \beta \in \mathbb{R}^C$ are the affine transformation parameters and $\mathbf{y} \in \mathbb{R}^{N \times C \times H \times W}$ is the output of BIN. We constrain the elements in $\rho$ to be in the range $[0, 1]$ simply by imposing bounds at the parameter update step:

$$
\rho \leftarrow \text{clip}_{[0,1]} \left( \rho - \eta \Delta \rho \right),
\tag{4}
$$

where $\eta$ is the learning rate and $\Delta \rho$ indicates the parameter update vector (*e.g.*, the gradient) determined by the optimizer. Intuitively, $\rho$ can be interpreted as a gate vector which decides whether to maintain or discard the style variation for each channel. If the style encoded by a channel is important to the task, the gate value increases toward 1 and the style is preserved by BN. If a style is unnecessary or disturbs the task, on the other hand, the corresponding gate approaches 0 to normalize the style through IN.

In practice, we found that it is beneficial to increase the learning rate for updating $\rho$. It fairly matches the theoretical aspect that the gradient of loss $\ell$ with respect to $\rho$ tends to be small because it is

Table 1: Top-1 accuracy (%) on CIFAR-10/100 (test set) and ImageNet (validation set) evaluated with ResNet-110 and ResNet-18, respectively. The results on CIFAR-10/100 are reported as the average and the 95% confidence interval over 10 repetitions.

|       | CIFAR-10           | CIFAR-100          | ImageNet |
|-------|--------------------|--------------------|----------|
| BN    | $93.72 \pm 0.18$   | $74.26 \pm 0.33$   | 69.89    |
| BIN   | $\mathbf{94.29 \pm 0.09}$ | $\mathbf{75.88 \pm 0.30}$ | **70.68** |

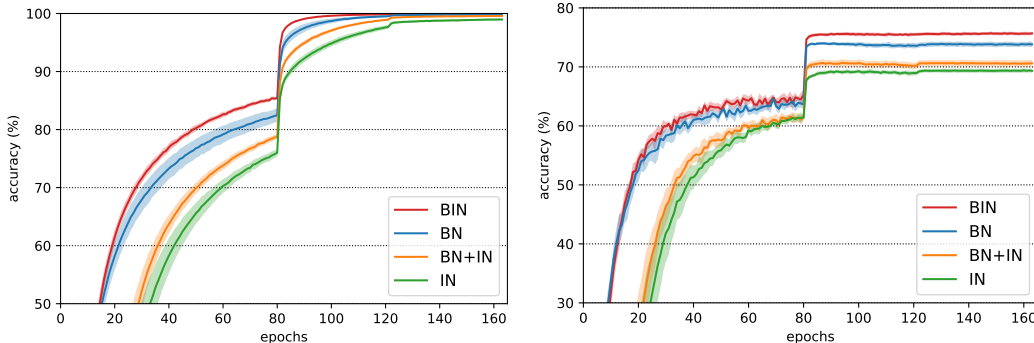

Figure 1: Training curves on CIFAR-100. We compare top-1 training accuracy (left) and testing accuracy (right) of ResNet-110 with different normalization methods. The solid lines and shaded areas represent the average and the 95% confidence interval over 10 repetitions, respectively.

calculated by multiplying the difference between $\hat{\mathbf{x}}^{(B)}$ and $\hat{\mathbf{x}}^{(I)}$:

$$\frac{\partial \ell}{\partial \rho_c} = \gamma_c \sum_N \sum_H \sum_W \left( \hat{x}_{nchw}^{(B)} - \hat{x}_{nchw}^{(I)} \right) \frac{\partial \ell}{\partial y_{nchw}}, \tag{5}$$

where $y_{nchw}$ is the $nchw$-th element of the output $\mathbf{y}$. Since the difference $\hat{\mathbf{x}}^{(B)} - \hat{\mathbf{x}}^{(I)}$ has small values especially when the style variation in the minibatch is marginal, training $\rho$ is facilitated by amplifying the difference using a larger learning rate.

## 4 Experiments

We demonstrate the effectiveness of BIN across a wide variety of scenarios in three different applications: object classification, multi-domain learning, and image style transfer. Ancillary experimental results on character recognition are also provided in the supplementary material. Throughout all experiments, the style gates ($\rho$) are initialized to 1 and trained with a learning rate multiplied by 10. Weight decay is not applied to the gate parameters.

### 4.1 Object Classification

We first evaluate BIN for general object classification using CIFAR-10/100 [15] and ImageNet [18] datasets. Deep Residual Networks [7]—ResNet-110 (32×32 input) on CIFAR-10/100 and ResNet-18 (224×224 input) on ImageNet—are employed as base architectures, which involve a BN layer after every convolutional layer. To evaluate BIN, we simply switch the BN to BIN while keeping all other parts unchanged. The networks are trained by SGD with a batch size of 128, an initial learning rate of 0.1, a momentum of 0.9, and a weight decay of 0.0001. On CIFAR datasets, we train the networks for 64K iterations, where the learning rate is divided by 10 at 32K and 48K iterations; on ImageNet, training runs for 90 epochs, where the learning rate is divided by 10 at 30 and 60 epochs.

**Comparison of normalization methods.** Table 1 compares the classification results, where BIN outperforms BN by considerable margins in all three datasets. The networks with BN and BIN use

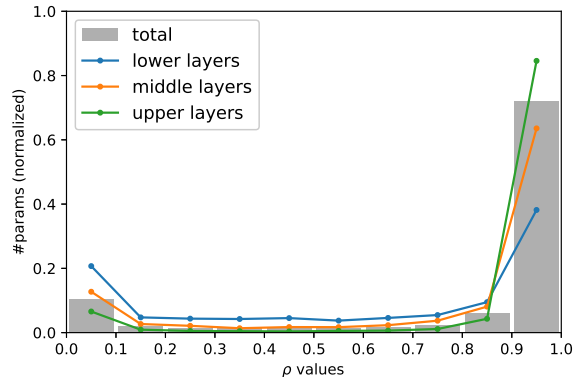

Figure 2: Distributions of the style gate parameters ($\rho$) of BIN in object classification. The lower, middle, and upper layers correspond to the three super-layers (*i.e.*, `layer1`, `layer2`, `layer3`) of ResNet-110 [7] configured as `conv1-bn1-relu-layer1-layer2-layer3-avgpool-fc`.

Table 2: Top-1 accuracy (%) on CIFAR-100 with various network architectures. We follow each author's notation for specifying the network configuration (*e.g.*, WRN-[depth]-[widening factor]).

|     | AlexNet | VGG-19 | ResNet-56 | ResNet-110 |
|-----|---------|--------|-----------|------------|
| BN  | 50.62   | 72.29  | 72.92     | 74.26      |
| BIN | **51.00** | **72.50** | **75.05** | **75.88** |

|     | PreResNet-110 | WRN-28-10 | ResNeXt-29, 8×64d | DenseNet-BC (L100, k12) |
|-----|---------------|-----------|-------------------|-------------------------|
| BN  | 76.49         | 80.92     | 80.50             | 76.93                   |
| BIN | **77.84**     | **81.48** | **81.57**         | **77.80**               |

almost the same number of parameters (*e.g.*, 1.73M and 1.74M for ResNet-110), which implies that the performance gain is not merely a consequence of increasing model capacity. The training curves on CIFAR-100 with varying normalization methods are also illustrated in Figure 1. BIN is compared with not only BN, but also IN and BN+IN—a naïve ensemble of BN and IN (*i.e.*, the $\rho$ in BIN is fixed to 0.5). We observe that replacing BN with IN deteriorates the performance significantly as reported in the literature. BN+IN somewhat reduces the gap between BN and IN, but is still far behind using BN only. On the other hand, BIN achieves higher training and testing accuracy than BN, as well as allows the network to train faster. The comparison between BIN and BN+IN verifies that the benefit of BIN does not just come from an ensemble effect of BN and IN. Instead, it comes from BIN's ability to learn to balance BN and IN so as to selectively normalize the styles irrelevant to class labels.

**Analysis on gate parameters.** To further understand the behavior of BIN, we investigate the learned values of the style gate parameters in BIN. Figure 2 shows the histogram of $\rho$ in the ResNet-110 trained on CIFAR-100. It forms a bimodal distribution where most of the parameters are biased toward 0 or 1, which represents that BIN tends to select either IN or BN rather than somewhere in between. In addition, the gate values close to 1 (using BN) occupy a much greater fraction than the values close to 0 (using IN). It conforms to the intuition that BN plays a major role in classification while IN plays a subsidiary role of reducing unnecessary style variations. Figure 2 also compares the gate distributions at different layers of the network. Lower layers spend relatively more parameters to IN (close to 0) than upper layers, because the styles—*i.e.*, global summary—of low-level features (*e.g.*, brightness, contrast, simple textures) are often irrelevant to the object classes. On the other hand, upper layers substantially rely on BN because they encode highly abstracted features that are closely related to object classes, of which the styles are as important as the shapes.

Table 3: Mixed-domain classification accuracy (%) of ResNet-18 on the Office-Home dataset averaged over 5-fold cross validation. The network is trained on the entire dataset and tested on each domain separately.

|      | Art   | Clipart | Product | Real-World | Avg.  |
|------|-------|---------|---------|------------|-------|
| BN   | 70.04 | 76.93   | 88.47   | 80.38      | 78.95 |
| BIN  | **72.23** | **77.27** | **89.12** | **81.68** | **80.08** |

Table 4: Domain adaptation accuracy (%) of DANN [4] (based on ResNet-18) on the Office-Home dataset. X→Y indicates X is the source domain and Y is the target domain.

|      | Ar→Cl | Ar→Pr | Ar→Rw | Cl→Ar | Cl→Pr | Cl→Rw |
|------|-------|-------|-------|-------|-------|-------|
| BN   | 37.23 | 46.17 | **63.30** | 49.38 | 50.90 | 60.09 |
| BIN  | **37.46** | **46.73** | 63.19 | **50.00** | **52.03** | **60.55** |

|      | Pr→Ar | Pr→Cl | Pr→Rw | Rw→Ar | Rw→Cl | Rw→Pr |
|------|-------|-------|-------|-------|-------|-------|
| BN   | 41.98 | 37.34 | 65.71 | 58.64 | 44.56 | 71.17 |
| BIN  | **43.00** | **37.69** | **65.94** | **59.67** | **45.02** | **71.85** |

**Scalability to different architectures.** We also evaluate the scalability of BIN by applying it to a wide range of CNN architectures. We borrow a publicly available implementation[2] of classification networks and experiment with every model in the repository—AlexNet [16], VGGNet [20][3], ResNet [7], PreResNet [8], WRN [26], ResNeXt [25], and DenseNet [9]. We construct two variants of each architecture using BN or BIN, train them on CIFAR-100 exactly following the hyper-parameters provided in the repository, and compare their testing accuracies. As shown in Table 2, BIN improves the performance across all tested architectures, which verifies the scalability of BIN with respect to model variety. It also suggests that BIN brings a distinct benefit that is not simply substituted by increased model capacity or elaborated network configuration.

## 4.2 Multi-Domain Learning

Next we apply BIN to multi-domain tasks to validate the advantage of BIN in handling the apparent style discrepancy between different domains. We employ the Office-Home dataset [23] which is recently introduced for evaluating domain adaptation with deep learning. It consists of 4 different domains—Art (Ar), Clipart (Cl), Product (Pr), and Real-World (Rw)—exhibiting substantial style variations. Each domain contains images from 65 categories shared across domains and the entire dataset comprises 15,588 images.

**Mixed-domain classification.** We first investigate mixed-domain classification where images from heterogeneous domains are mixed in the training data. We construct a training set by combining the 4 domains to train a domain-agnostic classification network, and evaluate the model on each domain separately. As done in the ImageNet classification in Section 4.1, we adopt ResNet-18 as a base architecture, and follow the same training policy except that the the network is pretrained on ImageNet and the initial learning rate is set to 0.01. We perform 5-fold cross validation and report the average accuracy in Table 3. BIN noticeably improves the performance on all of the 4 domains, which demonstrates that BIN effectively alleviates the style disparity between domains. It is especially effective in the Art domain which contains the fewest number of images (2,427 images), where exploiting other domains to learn style-invariant features is more valuable.

**Domain adaptation.** We additionally explore the domain adaptation problem involving a shift between training (source) and test (target) domains. We randomly split each domain into training and

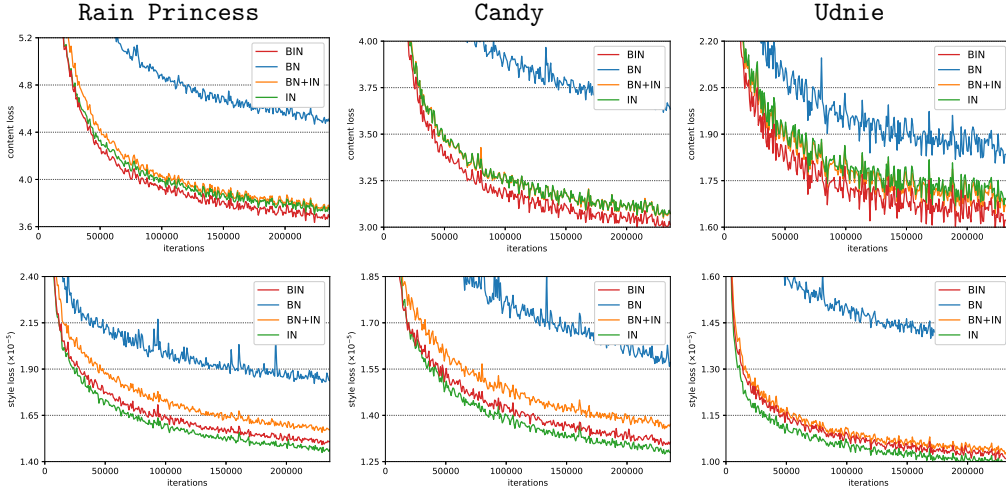

Figure 3: Training curves of style transfer networks with different styles and normalization methods. We compare the content loss (top row) and style loss (bottom row) averaged over 100 iterations..

test sets containing 80% and 20% of the images, respectively. Then we follow the "fully-transductive" protocol, where the training data from the source and the target domains are incorporated with and without class labels, respectively, which is also called as unsupervised domain adaptation. We employ Domain Adversarial Neural Network (DANN) [4] as a base algorithm, which is a recently proposed domain adaptation framework based on deep adversarial learning. It consists of three components trained end-to-end: a feature extractor to learn features shared across domains, a label predictor to classify object labels, and a domain classifier to make the features invariant to the domain shift. We reimplement the algorithm by constructing the feature extractor based on ResNet-18[4] where a 256-dimensional bottleneck is attached after the final average pooling layer. Each of the label predictor and domain classifier consists of two fully connected layers with a 1024-dimensional hidden layer ($256 \rightarrow 1024 \rightarrow 65$ and $256 \rightarrow 1024 \rightarrow 2$). We follow the training hyper-parameters provided by the author.

The test accuracies on the 12 transfer tasks in the Office-Home dataset are presented in Table 4. Although not by significant margins, BIN consistently surpasses BN on 11 out of the 12 tasks. In this scenario BIN identifies and normalizes the styles associated with domain labels as well as not associated with class labels, allowing the features to be more invariant to domain changes. In addition, noticing that the target domain is trained without class labels, BIN potentially benefits addressing problems lacking labeled data by learning more generalizable features from another domain.

### 4.3 Image Style Transfer

Finally we conduct experiments on style transfer where IN is more appropriate than BN, in order to confirm that BIN also retains the benefit of IN. We adopt a feed forward style transfer algorithm [14] which consists of an image transformation network and a loss network for calculating the content loss and style loss. We train the image transformation network by switching normalization layers, using content images from the MS-COCO dataset [17] following the same training policy as in [14].

**Comparison of normalization methods.**   Figure 3 compares the training curves with different normalization methods. As reported in [21, 22, 10], IN brings significant improvement over BN by discarding style information from content images. A naïve ensemble of BN and IN (BN+IN) produces a comparable content loss with IN but degenerates the style loss with a considerable margin because disturbing styles of content images are still remained. In comparison, BIN achieves slightly lower content loss than IN by preserving important styles—that might be relevant to the contents—from

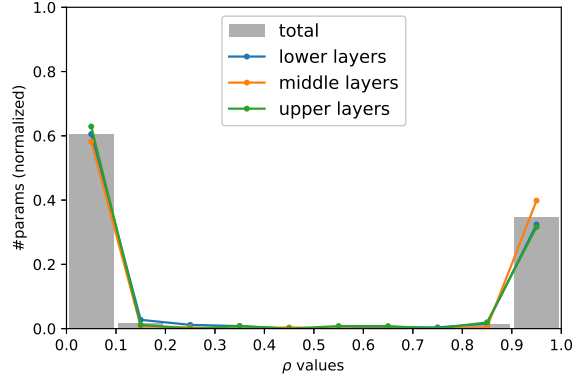

Figure 4: Distributions of the style gate parameters ($\rho$) of BIN in image style transfer (`Rain Princess`). The lower, middle, and upper layers correspond to the first, third, and last residual blocks, respectively, of the image transformation network [14] containing five residual blocks.

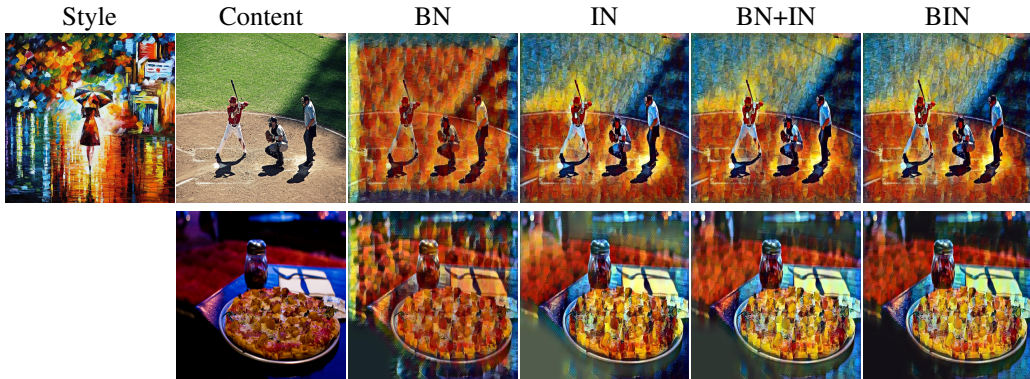

Figure 5: Example results of style transfer with different normalization methods. BIN mostly produces similar results to IN or BN+IN (top), but sometimes generates improved results than the others by selectively preserving the style of the content image (bottom).

content images. Although it produces marginally higher style loss than IN due to the remaining styles of content images, the gap is clearly mitigated compared with BN+IN because it selectively discards impeditive styles only.

**Analysis on gate parameters.** As done in Section 4.1, we perform analysis on the gate parameters of BIN. Figure 4 represents the histogram of $\rho$ in the trained image transformation network, which forms a bimodal distribution as in Figure 2. However, the fraction of gates close to 0 is much higher (even though they are initialized to 1) compared to Figure 2. This not only ensures that IN is more suited to a style transfer task than BN, but also demonstrates that BIN exhibits completely different behavior depending on the task. Furthermore, the distributions at different layers show almost no difference unlike Figure 2, because the depth of the transformation network (image→image) is less related with the level of abstraction than the depth of the classification network (image→class label).

**Qualitative comparison.** Some qualitative examples of style transfer are illustrated in Figure 5. While BN leads to inferior stylization quality as reported in [21, 22], the other options (IN, BN+IN, and BIN) produce better results with similar quality in most cases (first row). The distinct behavior of BIN is observed when the content image contains a meaningful style (second row). BIN tends to preserve the style of a content image to some degree as well as successfully transfers the target style, while IN and BN+IN lose the original style almost completely. More results are also provided in the supplementary material.

# 5  Conclusion

In this paper we propose Batch-Instance Normalization (BIN) to adaptively learn to normalize style information from images. It binds BN and IN leveraging the gate parameters to determine the proper normalization method based on the importance of style features. We demonstrate that simply replacing BN with BIN considerably improves the performance of BN-based models in various image recognition tasks. BIN is also successfully applied to image stylization as an alternative to IN, suggesting the flexibility of BIN for different purposes.

Ever since the breakthrough achieved by batch normalization, a number of studies have suggested improved normalization techniques (batch renormalization [12], weight normalization [19], layer normalization [1], group normalization [24], etc.). They mostly focus on addressing the minibatch dependencies of BN in constrained situations, but still none of them is as generalizable as BN to a wide range of applications. We hope our work sheds light on another aspect of improving normalization, identifying and removing unnecessary variations from data, which might be the fundamental role of normalization.

## Footnotes

[1]https://github.com/hyeonseob-nam/Batch-Instance-Normalization

[2]https://github.com/bearpaw/pytorch-classification

[3]Since AlexNet and VGGNet do not contain BN layers, we modify the networks by adding a BN (or BIN) layer after every convolutional layer.

[4]DANN originally uses AlexNet, but we replace it by ResNet-18 for natural incorporation of normalization layers and comparability with above experiments.

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
