[Supplementary Material · supp.pdf]

# Supplementary Material

## A More Examples of Image Style Transfer

We provide more qualitative results of style transfer to supplement the Section 4.3 in the main submission. Figure 1 presents the cases where BIN produces visually similar results with IN and BN+IN; Figure 2 illustrates the cases where BIN performs better than other methods by partially preserving the style information of content images.

## B Experiments on Character Recognition

We performed subsidiary experiments on character recognition datasets—ICDAR2003 characters [3], ICDAR2005 (digits, lower and upper case characters) [2], and Chars74K [1]—involving substantial style variations. We build relatively shallow residual networks with BN or BIN and train them following the same hyper-parameters as in the Section 4.1, except that training runs for 150 epochs where the learning rate is decreased at 50 and 100 epochs. BIN consistently outperforms BN in all experiments as shown in Table 1.

Table 1: Top-1 accuracy (%) on various character recognition datasets. We repeat each experiment 5 times and report the average accuracy with the 95% confidence interval.

|  |  | ResNet-8 | ResNet-14 | ResNet-20 | ResNet-32 |
|---|---|---|---|---|---|
| ICDAR2003 | BN | $64.98 \pm 0.63$ | $68.27 \pm 0.54$ | $69.32 \pm 0.71$ | $69.94 \pm 0.49$ |
|  | BIN | $\mathbf{68.55 \pm 0.32}$ | $\mathbf{70.78 \pm 0.50}$ | $\mathbf{71.48 \pm 0.27}$ | $\mathbf{71.83 \pm 0.25}$ |
| ICDAR2005 (Digits) | BN | $58.53 \pm 1.52$ | $73.15 \pm 1.83$ | $77.26 \pm 1.53$ | $78.58 \pm 1.41$ |
|  | BIN | $\mathbf{64.52 \pm 2.44}$ | $\mathbf{76.45 \pm 1.50}$ | $\mathbf{78.78 \pm 1.38}$ | $\mathbf{81.62 \pm 1.90}$ |
| ICDAR2005 (Lower) | BN | $83.58 \pm 0.80$ | $87.70 \pm 0.47$ | $89.42 \pm 0.78$ | $89.14 \pm 1.07$ |
|  | BIN | $\mathbf{84.22 \pm 0.78}$ | $\mathbf{89.20 \pm 1.01}$ | $\mathbf{90.35 \pm 0.60}$ | $\mathbf{90.26 \pm 0.52}$ |
| ICDAR2005 (Upper) | BN | $83.78 \pm 0.84$ | $87.54 \pm 0.89$ | $88.75 \pm 1.29$ | $89.71 \pm 0.73$ |
|  | BIN | $\mathbf{85.76 \pm 0.67}$ | $\mathbf{89.05 \pm 0.62}$ | $\mathbf{89.43 \pm 0.95}$ | $\mathbf{90.62 \pm 0.55}$ |
| Chars74K | BN | $72.13 \pm 0.85$ | $76.80 \pm 0.50$ | $77.29 \pm 0.68$ | $78.86 \pm 0.49$ |
|  | BIN | $\mathbf{74.77 \pm 0.47}$ | $\mathbf{78.11 \pm 0.52}$ | $\mathbf{78.39 \pm 0.45}$ | $\mathbf{79.18 \pm 0.34}$ |

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

Figure 1: Style transfer examples where BIN produces similar results with IN and BN+IN.

Figure 2: Style transfer examples where BIN produces distinct results from the other methods. BIN tends to preserve the style of a content image to some degree, especially showing effectiveness when the image contains dark areas.