[Reviews · NeurIPS 2018]

Reviewer 1



The submission introduces a new normalisation layer, that learns to linearly interpolate between Batch Normalisation (BN) and Instance Normalisation (IN). The effect of the layer is evaluated on object recognition benchmarks (CIFAR10/100, ImageNet, multi domain Office-Home) and Image Style Transfer. For object recognition it is shown that for a large number of models and datasets the proposed normalisation layer leads to a small but consistent improvement in object recognition performance. In Image Style Transfer a systematic difference in performance is more difficult to detect, but examples are shown in which using the new normalisation layer leads to the ability to preserve parts of the content image better (e.g. dark backgrounds). A nice analysis is performed on the relative weighting of IN and BN in different tasks. First it is shown that the interpolation weights follow a bi-modal distribution. Thus features either use IN to become invariant or BN to retain input variation across images. Moreover, while for object recognition the majority of features use BN, the fraction of IN usage increases substantially for style transfer as expected from the requirements of the task. Nevertheless, I have a couple of concerns/questions. 
1.) A main concern is that the effect size is often quite small and no analysis is provided how this compares to the variation in performance from running multiple random initialisations. Although the consistency of the improvement over models and datasets makes me confident that there is an effect, I would like to see at least for one model a statistical analysis giving error estimates and evaluating significance of the improvement. Preferably for the results in table 1. 2.) The results should be compared to the Adaptive Normalisation proposed in [1] which learns a linear interpolation between BN and the identity function. Ideally I would like to see 2 more conditions to compare with: one linearly interpolating between IN and identity and one interpolating between BN and identity. Overall the submission describes a simple change to existing architectures that appears to have a consistent positive effect on model performance for several models and datasets, which can be of interest to the larger Deep Learning community at NIPS. If the authors can address my questions in the rebuttal, I am happy to increase the score to 7. Edit: Based on the authors' response showing statistical significance of the improvement and comparison to BN-Identity, I increase the score to 7 and recommend acceptance of the submission. References [1] Fast image processing with fully-convolutional networks
 Q Chen, J Xu, V Koltun - IEEE International Conference on Computer Vision, 2017

Reviewer 2



Update following the author rebuttal: I would like to thank the authors for providing a thoughtful rebuttal and addressing the points I raised in my review. ----- The paper proposes a method -- termed Batch-Instance Normalization (BIN) -- for automatically learning which of batch (BN) or instance (IN) normalization should be employed for individual feature maps in a convolutional neural network. The justification is simple and intuitive: in certain cases it is beneficial to standardize features on a per-example basis (e.g., in most cases global illumination is irrelevant for object classification), but in other cases doing so is hurtful (e.g., when predicting weather or time global illumination is a very informative feature). The authors propose a simple idea: batch normalization and instance normalization are blended through a convex combination of both forms of normalization, and the blending coefficient is treated as a model parameter and learned jointly with the rest of the network parameters. I like the simplicity of the proposed method. The paper is well-written, straightforward, and contextualizes the idea well alongside relevant work. The authors present experimental results for object classification, multi-domain training, and artistic style transfer. For object classification, accuracies and learning curves are presented for ResNet architectures trained on CIFAR10, CIFAR100, and ImageNet, showing that BIN consistently outperforms BN, IN, and BN+IN (i.e., BIN with a constant blending coefficient of 0.5) by a small margin. In particular, the comparison with BN+IN suggests that the adaptive nature of the proposed idea is responsible for the observed improvements, rather than the the ensembling of BN and IN. The authors also analyze the distribution of blending coefficients and show that the distribution is roughly bimodal, with most of the probability mass concentrated near the BN end of the spectrum and a smaller amount concentrated near the IN end of the spectrum, which aligns well with the intuition put forward by the paper. One criticism I have to voice is that the absence of confidence intervals for the presented accuracies makes it hard for me to evaluate whether the improvements presented are statistically significant in some cases. The consistent nature of the improvements -- best exemplified by Table 2 -- goes some way towards building confidence in their statistical significance, but I would nevertheless like to have a sense of what sort of variability to expect due to simple factors like random weight initialization and whether the observed improvements deviate significantly from it. For multi-domain training, the authors compare BIN to BN in a mixed-domain classification setting. Once again, consistent improvements are shown, and the authors conclude that BIN’s ability to alleviate style disparities between domains is responsible for the increase in accuracies. I believe this is a likely explanation, but I would like to see whether BIN also improves performance when training on individual domains, as it could be that the ability to alleviate intra-domain style disparities is also helpful. The authors also compare BIN to BN in a domain-adaptation setting and show that BIN outperforms BN in most settings by a small margin. For style transfer, the paper compares BIN with BN, IN, and BN+IN when training a feed-forward style transfer network on a single style image. They conclude that BIN makes a trade-off between the content loss and the style loss (which are found to be respectively slightly lower and slightly higher than their BN counterparts), and that BIN is better than BN+IN at minimizing both the content and style losses. Presenting results on more than one style would help build confidence in the significance of the observed differences; alternatively, presenting results for a multi-style feedforward network like the ones proposed by Dumoulin et al., Huang et al., or Ghiasi et al. (G. Ghiasi, H. Lee, M. Kudlur, V. Dumoulin, J. Shlens. “Exploring the structure of a real-time, arbitrary neural artistic stylization network”. Proceedings of the British Machine Vision Conference (2017).) would be equally convincing. In summary, the paper proposes a simple idea, explains it in a straightforward and well-written fashion, and presents thorough and convincing empirical evidence that it improves performance in multiple problem settings. The biggest criticism I have to voice is that I would like to see confidence intervals for the accuracies. Although the idea itself is not groundbreaking, I believe it is a useful and relevant addition to a deep learning researcher’s toolkit.

Reviewer 3



I have changed by score to 6 after reading the rebuttal and other reviews. I highly like the main point of this paper, that it is helpful to selectively normalize some styles. I agree that simplicity is not a bad thing if the algorithm indeed performs well, but I still think the margin of improvement (both in ImageNet classification and style transfer) is not very impressive. So I will not give it a higher score. ==================================== This paper proposes a simple normalization layer that linearly combines the IN output and BN output, named BIN. Experiments show BIN performs better than BN and IN in terms of image classification and style transfer. Pros: 1. The paper is well-written and easy to read. 2. The interpretation that "BIN learns to selectively normalize only disturbing styles while preserving useful styles" makes sense to me. It is also supported by the fact that earlier layers tend to use IN more while higher layers tend to use BN. 3. It is shown that BIN generalizes well to different architectures and different tasks. Cons: 1. The proposed method is quite straightforward - adding a gate to switch between BN and IN. The technical novelty is limited. 2. BIN introduces extra computational cost compared with BN or IN. Although I expect the difference to be minimal, the authors should mention it somewhere. Overall I think this is an interesting paper that shows empirical improvement over baselines. However, my main concern is its technical depth seems low.